# Hospital-Based Influenza and Pneumococcal Vaccination for Cancer Patients on Active Treatment and Their Family Members during the COVID-19 Pandemic in Italy: A Single-Center Experience

**DOI:** 10.3390/vaccines12060642

**Published:** 2024-06-08

**Authors:** Davide Dalu, Anna Lisa Ridolfo, Lorenzo Ruggieri, Maria Silvia Cona, Agostino Riva, Davide De Francesco, Chiara Tricella, Cinzia Fasola, Sabrina Ferrario, Anna Gambaro, Benedetta Lombardi Stocchetti, Valeria Smiroldo, Gaia Rebecchi, Sheila Piva, Giorgia Carrozzo, Spinello Antinori, Nicla La Verde

**Affiliations:** 1Department of Oncology, Luigi Sacco Hospital, ASST Fatebenefratelli Sacco, 20157 Milan, Italy; dalu.davide@asst-fbf-sacco.it (D.D.); cona.silvia@asst-fbf-sacco.it (M.S.C.); chiara.tricella@unimi.it (C.T.); fasola.cinzia@asst-fbf-sacco.it (C.F.); ferrario.sabrina@asst-fbf-sacco.it (S.F.); gambaro.anna@asst-fbf-sacco.it (A.G.); benedetta.lombardi@unimi.it (B.L.S.); vsmiroldo@asst-rhodense.it (V.S.); gaia.rebecchi@unimi.it (G.R.); nicla.laverde@asst-fbf-sacco.it (N.L.V.); 2Department of Infectious Diseases, Luigi Sacco Hospital, ASST Fatebenefratelli Sacco, 20157 Milan, Italy; ridolfo.annalisa@asst-fbf-sacco.it (A.L.R.); riva.agostino@asst-fbf-sacco.it (A.R.); carrozzo.giorgia@asst-fbf-sacco.it (G.C.); antinori.spinello@asst-fbf-sacco.it (S.A.); 3Luigi Sacco Department of Biomedical and Clinical Sciences DIBIC, University of Milan, 20157 Milan, Italy; 4School of Medicine, Stanford University, Stanford, CA 94305, USA; d.defrancesco84@gmail.com; 5Department of Oncology, Fatebenefratelli Hospital, ASST Fatebenefratelli Sacco, 20121 Milan, Italy; sheila.piva@asst-fbf-sacco.it

**Keywords:** hospital vaccination, influenza, pneumococcal, cancer

## Abstract

In patients with cancer, tumor- and treatment-induced immunosuppression are responsible for a four-fold increase in morbidity and mortality caused by influenza and invasive *Streptococcus pneumoniae* infections compared to the general population. The main oncology societies strongly recommend vaccination in patients with cancer to prevent these infections. However, vaccine hesitancy is a main concern in this population. The aim of this study was to assess the feasibility of in-hospital vaccination for patients under anticancer treatment and their family members (FMs) against influenza and pneumococcal infections during the COVID-19 pandemic in order to increase vaccine coverage. This was a single-center, prospective, observational study conducted at the Department of Oncology of Luigi Sacco University Hospital (Milan, Italy) between October 2020 and April 2021. The main primary outcome was the incidence of influenza-like illness (ILI) and pneumococcal infections. The main secondary outcome was safety. A total of 341 subjects were enrolled, including 194 patients with cancer and 147 FMs. The incidence of ILI was higher among patients than among FMs (9% vs. 2.7%, OR 3.92, *p* = 0.02). Moreover, two subjects were diagnosed with pneumococcal pneumonia. The most frequent vaccine-related AEs were pain in the injection site (31%) and fatigue (8.7%). In conclusion, this hospital-based vaccination strategy was feasible during the COVID-19 pandemic, representing a potential model to maximize vaccine coverage during a public health emergency.

## 1. Introduction

Influenza is an acute infection of the respiratory tract caused by RNA viruses belonging to the family Orthomyxoviridae. Three types of influenza virus (A, B, and C) can infect humans, but only influenza virus types A and B are associated with seasonal epidemics [1,2,3]. During the 2019–2020 season in Italy, more than 5 million cases of influenza were recorded in the general population [4].

In patients with cancer, tumor- and treatment-induced immunosuppression lead to up to a four-fold increase in morbidity and mortality caused by influenza compared to the general population. Indeed, previous studies reported a higher incidence of complications in patients with cancer, such as bacterial superinfections and respiratory failure, and higher rates of hospitalization and mortality [5,6,7,8,9].

Pneumococcal disease includes different infections caused by *Streptococcus pneumoniae*, a capsulated, aerobic, and alpha-hemolytic Gram-positive diplococcus. In general, pneumococcal infection can cause invasive pneumococcal disease (IPD, with bacteremia) or non-invasive pneumococcal disease (otitis, sinusitis, pneumonia, without bacteremia) [10]. In 2019, the incidence of pneumococcal infections (PI) in Italy was 2.5 per 100,000 people; the most frequent clinical presentation was sepsis/bacteremia (34–39%) followed by pneumonia (33–36%) and the mortality rate was higher in the elderly and in immunocompromised subjects [11]. The incidence of IPD in patients with solid and hematological malignancies is 4 to 62 times higher than the general population [12].

Vaccinations are the current recommended standard of prevention for both influenza and pneumococcal infections [13,14,15]. Unfortunately, limited data are available on the safety, immunogenicity, and efficacy of influenza and pneumococcal vaccines in cancer patients undergoing systemic treatment [16,17,18,19,20,21,22,23,24,25,26,27,28,29]. Additionally, the optimal timing of vaccination in relation to the administration of anticancer therapy is still unclear [24].

Nevertheless, the Italian Ministry of Health (National Vaccine Prevention Plan 2017–2019) and the main oncology societies include cancer patients among the most frail and vulnerable population that needs to be vaccinated with both the seasonal influenza vaccine and the sequential PCV13 (pneumococcal conjugate vaccine against 13 serotypes of bacteria) and PPSV23 (pneumococcal polysaccharide vaccine against 23 serotypes of bacteria). The influenza vaccine is also recommended for all family members (FMs) of patients with cancer [30,31,32] in order to boost this effect. Despite this, adherence to influenza and pneumococcal vaccines is scarce in this population [33]. Currently, hospital-based vaccination strategies are scarcely employed. Indeed, the vaccination of patients with cancer is demanded by community health services and recruitment relies mainly on individual physician advice. For these reasons, real-world studies that evaluate the efficacy of proactive hospital-based vaccination of patients with cancer and their FMs are warranted. Indeed, the aim of this study was to assess the feasibility of in-hospital vaccination of patients under anticancer treatment and their family members (FMs) against influenza and pneumococcal infections during the COVID-19 pandemic as a part of the continuum of cancer care, with the purpose of increasing vaccine coverage.

## 2. Materials and Methods

### 2.1. Study Design

This was a single-center, prospective, observational study conducted at the outpatient clinic of the Department of Oncology of Luigi Sacco University Hospital in Milan, Italy, between October 2020 and April 2021. 

We enrolled unselected consecutive cancer patients who were under active anticancer treatment and their FMs, defined as all the subjects related by blood or marriage that lived in the same household or had frequent contacts with patients. 

Vaccination was a single dose of quadrivalent inactivated influenza vaccine and a single dose of pneumococcal vaccine, as recommended by the World Health Organization (WHO) and by the Italian Ministry of Health [34]. 

The two vaccines were injected into the deltoid muscle of two different arms in the same session. 

During daily visits, the oncologist proposed the vaccinations to patients and their FMs and provided all of the information about safety and efficacy. After patient acceptance, written informed consent was required. Since this approach was proven safe and effective in previous studies [6,24,35], departmental nurses scheduled the vaccinations on the date of the first anticancer treatment cycle to avoid unnecessary patient hospital visits. 

The vaccinations of the FMs were scheduled on the same day after the planned medical activities to avoid overcrowding of the waiting room. 

Nurses administered vaccines in a dedicated room of the outpatient clinic. Regarding influenza vaccination, quadrivalent inactivated high-dose (HD) vaccine for patients aged over 65 years (FLUZONE^®^ HD, Sanofi Pasteur, Swiftwater, PA, USA) and quadrivalent inactivated vaccine for patients aged 65 years old or less (VAXIGRIP Tetra^®^, Sanofi Pasteur, Swiftwater, PA, USA) [36] were administered. Furthermore, the pneumococcal vaccines employed were PCV13 (PREVENAR 13^®^, Pfizer, New York, NY, USA) and PPSV23 (PNEUMOVAX23^®^, MSD, Rahway, NJ, USA) [37]. 

As per WHO guidelines, immunocompetent FMs aged 65 years or more received a dose of PCV13 followed by a dose of PPSV23 at least one year afterward. For FMs who previously received PPSV23 when aged 65 years or less and for whom an additional dose of PPSV23 was indicated when aged 65 years or more, the subsequent PPSV23 dose was administered at least one year after PCV13 and at least 5 years after the most recent dose of PPSV23 [38]. The same sequential pneumococcal vaccinations were administered to patients, with a recommended interval between PCV13 and PPSV23 injections of at least 8 weeks. 

To simplify the vaccination protocol, to avoid additional visits, and to reduce visits to other health services during the pandemic period, we recommended to administer PPSV23 and the next influenza vaccination simultaneously. 

The study was conducted in accordance with the principles of the Declaration of Helsinki and the International Conference on Harmonization and Good Clinical Practice Guidelines. The local Ethics Committee (Milano Area A) approved the study procedures. All subjects provided written informed consent. 

Patients were eligible to participate if they were aged at least 18 years and were under active systemic treatment for hematological or solid cancers. FMs living permanently or temporarily in the same household as patients were eligible if they were aged at least 18 years. Both patients and FMs could have been excluded at the individual discretion of the attending physician if they were considered not eligible for vaccination for any clinical reason. 

Investigational outcomes were divided into co-primary and co-secondary outcomes. 

The main co-primary outcome was the incidence of ILI, defined as feverishness or a measured temperature of at least 37.7 °C (≥100 °F), plus cough or sore throat according to the CDC ILI surveillance definition [39], and the number of PIs. In case of ILI, a nasal swab and RT-PCR SARS-CoV-2 testing was performed to exclude SARS-CoV-2 infection. PI was defined as microbiologic confirmation of blood, sputum, or other lower respiratory tract sample trough positive S. pneumoniae Gram stain, cultures, or urinary antigen positive test. 

Other co-primary outcomes were the frequency of visits to the emergency department (ED), the number of hospital admissions (HA), and the delay of treatment for more than 7 days (DoT) related to ILI or PI. 

The main co-secondary outcome was to assess the adverse events (AEs) related to influenza and pneumococcal vaccines. Local AEs were defined as soreness, redness, or swelling in the site of injection. Systemic AEs were defined as headache, fever, nausea, muscle aches, fatigue, dizziness, and decreased appetite. AEs occurring within 3 days after vaccination were registered after telephone call or hospital visit. 

Other co-secondary outcomes were the association between individual variables with the occurrence of AEs and the differences between patients and FMs in the evaluated outcomes. 

Sociodemographic information, including gender and age, was actively collected from patients and their FMs. Race and ethnicity were not recorded since more than 95% of subjects were Caucasian. 

Additionally, medical information, such as cancer type and stage, number of previous lines of systemic treatment, schedule of administration, use of steroids, and the Charlson Comorbidity Index (CCI) [40], was collected from electronic medical records. 

Investigational variables were extracted through telephone interviews and from medical records. Monthly monitoring of patients and FMs was scheduled to register investigational variables, such as ILI, PI, ED, HA, and DoT. 

Follow-up ended 6 months after the administration of the vaccinations. A specific questionnaire was to be completed 4 weeks after vaccination in the case that AEs occurred in both patients and their FMs within 3 days after vaccination.

### 2.2. Statistical Analysis

All investigational variables were summarized as frequencies and proportions or as medians and interquartile range (IQR); differences between groups (patients vs. FMs) were assessed for significance using the chi-square test (or Fisher’s exact test when appropriate) or the T-test (or the Wilcoxon test when appropriate). Descriptive statistics were employed to evaluate the frequency of ILI, associated symptoms, interventions, and general patient information. Given the non-randomized design of the study, a multivariable logistic regression model was used to adjust for age and gender.

## 3. Results

### 3.1. Patient Demographics and Clinical Characteristics 

During the study period, 341 subjects were enrolled, including 194 patients and 147 FMs. Participation was offered to 277 patients, but 83 refused (adherence: 70%). The baseline characteristics according to subgroups are described in Table 1.

The median age of patients was 63 years (IQR 51, 73) and 131 (67.5%) were female. The primary tumor site was the breast in 88 patients (45.5%), genitourinary (GU) in 36 (18.5%), gastrointestinal (GI) tract in 31 (16%), lung in 20 (10.3%), hematological malignancy in 14 (7.2%), and head and neck in 4 (2.1%). The majority of patients had an advance stage of disease (59%). During the study, 38% of patients were treated with chemotherapy, 31% with targeted therapy, 17% with combined chemotherapy and targeted therapy, and 14% with hormone therapy. A CCI of at least 5 was observed in 157 (83%) patients. The most common comorbidities were cardiovascular (39%), dysmetabolic (9%), broncho-pulmonary (7%), and other diseases (45%). 

The median age among FMs was 59 years (IQR 29–69) and 73 of them (49.7%) were female. All enrolled subjects received the influenza vaccine. The quadrivalent inactivated HD vaccine was administered to 45% of the patients and 34% of the FMs. Additionally, 47% of the patients and 56% of the FMs received the influenza vaccine for the first time. A total of 175 (90.2%) patients received the pneumococcal vaccine, among whom 171 received PVC13 and 4 received PPSV23. The rest refused to receive the vaccine. In the FM group, 69 (46.9%) individuals received the pneumococcal vaccine. The rest did not receive the vaccine since they were under the minimum age according to the main recommendations [2,15]. FMs suitable for vaccination received PCV13 only.

### 3.2. Investigational Outcomes

#### 3.2.1. Influenza-like Illness and Pneumococcal Infections

The incidence of ILI was 9.3% (18 cases) among patients and 2.7% (4 cases) among FMs (*p* = 0.03), respectively (Table 2).

None tested positive for SARS-CoV-2 PCR testing on nasal swab. The influenza test was not performed. All cases of ILI occurred more than 30 days after vaccination. The logistic regression analysis confirmed that the likelihood of ILI was significantly higher in patients than in FMs (odds ratio (OR) 3.92, *p* = 0.02). Moreover, 1 patient and 1 FM developed pneumococcal pneumonia. Both diagnoses were confirmed by urinary antigen test, but this test did not define the pneumococcal serotype. The patient was >65 years old and received the PCV13 vaccine more than 30 days before developing pneumonia. The FM was> 65 years old but refused pneumococcal vaccination.

Overall, 3 patients were admitted to the Emergency Unit and hospitalized due to severity of ILI, whereas none of the FMs were hospitalized. No cases of death related to progression of ILI were observed.

During the study period, treatment was delayed in 2 patients: in one case for ILI and in the other for COVID-19 requiring hospitalization.

#### 3.2.2. Adverse Events

Table 3 summarizes the frequency of AEs reported by study participants. Almost half of the participants (47.7%) reported at least one AE, with local reactions being the most frequent. The frequency of any local reaction was higher among patients than among FMs (34% vs. 21%, *p* = 0.02), and the logistic regression analysis showed significantly higher odds for patients compared to FMs (OR = 1.98, 95% confidence interval (CI95) 1.18–3.31, *p* < 0.01).

Of all reported local AEs to the influenza vaccine, the most frequent was pain in the site of injection (50 patients and 28 FMs). No difference in local AEs was observed with the administration of high- or standard-dose influenza vaccine.

Pain in the site of injection was also the most common AE to the pneumococcal vaccine (52 patients and 13 FMs). The overall frequency of local AEs was higher in patients compared to FMs (35.5% vs. 20.2%, *p* = 0.02), and the logistic regression analysis showed a significantly higher likelihood of local AEs for patients (OR (CI95) 1.90 (1.02–3.55), *p* = 0.04). Given the impossibility of distinguishing between systemic AEs caused by the influenza vaccine or by the pneumococcal vaccine, they were grouped together. Firstly, we calculated the rates of systemic AEs in patients and FMs, stratified for the vaccine administered (both pneumococcal and influenza vaccines vs. only influenza vaccine). The results are reported in Table 3. Subsequently, the rate of systemic AEs in patients who received both vaccines and in those who received only the influenza vaccine were compared. Systemic AEs were more frequent among patients who received both vaccines (20.6%) than in those who only received the influenza vaccine (5.3%), with a statistically significant difference even after adjusting for age and gender (*p* = 0.002, logistic regression OR (CI95) 6.8 (2.0–22.9). Overall, the most frequent AE was fatigue (8.7%).

#### 3.2.3. Investigational Outcomes According to Clinical Variables

There was no significant difference in the incidence of ILI among patients according to the type of systemic treatment, schedule of administration, number of previous lines of therapy, and CCI (see Appendix A). Furthermore, the use of steroids did not impact the likelihood of developing ILI (OR (CI95) 0.67 (0.24–1.87), *p* = 0.44) and it was not associated with higher odds of AEs and ED/HA (Table 3).

## 4. Discussion

Influenza and pneumococcal vaccinations are effective preventive therapies. Previous studies showed a higher risk of mortality for patients with cancer, which was substantially related to their immune suppression [41,42]. During the COVID-19 pandemic, influenza caused relevant disruptions in cancer diagnosis and treatment. Moreover, similarities in the clinical manifestation of influenza and COVID-19 forced symptomatic patients to undergo excessive preventive quarantine. Despite this, vaccine hesitancy remains a significant concern in cancer patients. In this study, half of the subjects had never received these vaccines in their lives. Furthermore, about 30% of eligible patients refused enrollment. Decision to postpone vaccine administration, preferred territorial vaccination, and individual rejection were the most frequent reasons. Indeed, adherence in this study was 70%, compared to 23.7% in 2020–2021 [43] and 16.8% in 2019–2020 [44] in Italy in the general population.

This study describes the results of an in-hospital vaccination campaign that aimed to increase vaccine coverage in cancer patients and their FMs during a crucial period, such as the COVID-19 pandemic. In this context, the most frequent barriers to vaccination were confidence (decreased perceived vaccine effectiveness, increased perceived risk of side effects), complacency (decreased worry about the risk and the severity of the disease), calculation (decreased perceived own and social benefit of the vaccine), and convenience (perceived of low self-efficacy, worry about not having a regular source of care). 

The results of our study showed that vaccinated cancer patients had about 3.4-fold increased risk of developing ILI compared to FMs. This was in line with the limitations of vaccine efficacy in cancer patients due to their immune-compromised status. Despite the lower immunogenicity of vaccines in patients with cancer, oncology societies recommend vaccination to reduce influenza and pneumococcal complications [7]. Moreover, cancer patients benefit from the protection against influenza provided by their vaccinated FMs and the consequent reduction in virus circulation within the household.

Previous studies on vaccine effectiveness in patients with cancer are lacking and evidence is derived from studies with small sample sizes [45]. The frequency of ILI in vaccinated cancer patients reported so far appears to be higher than that observed in this study (300–350 vs. 92.8 per 1000 people) [46]. This might be related to the protective role of FM vaccination. In addition, COVID-19 mitigation measures, such as social isolation, face masks, hand-washing, physical distancing, increased ventilation in indoor spaces, school closures, reduced traveling, strict protocols for accessing hospital services, and visiting arrangements in health care settings may have influenced this result [47]. 

Indeed, globally, influenza virus and *Streptococcus pneumoniae* infection incidence during the 2020–2021 season was unusually low. In Italy, compared to the last season (2018–2019) before the pandemic, the peak in ILI decreased from 14 to 1.5 per 1000 people and the incidence rate of non-invasive pneumococcal disease decreased from 2.19 to less than 1 case per 1000 people [48].

In this study, we detected a higher incidence of vaccine-related adverse local reactions in cancer patients as compared to their FMs. This result could have been altered by the simultaneous presence of adverse events related to anticancer treatment. On the other hand, no difference was observed in systemic AEs between patients and FMs. This difference could have been related to the anti-inflammatory effects of supportive corticosteroid treatment during anticancer treatment [49].

Furthermore, the results of our study suggest that some patients and treatment variables (corticosteroids, systemic treatment, schedule of administration, number of previous lines of therapy, comorbidity) did not influence the impact and safety of influenza and anti-pneumococcal vaccines.

Finally, currently, there is no agreement on the optimal timing for administering vaccines to patients with cancer undergoing active anticancer treatment. Our choice to standardize the administration of vaccines on the first day of the anticancer treatment cycle was mainly based on patient convenience (i.e., to avoid further visits). Moreover, this approach was supported by previous studies [6,35]. 

We believe that an in-hospital strategy could represent a useful model during a public health emergency because it ensures that patients will receive essential care, as vaccines, reducing the exposure to interhuman contagion by avoiding further transfers. Besides, it reduces the pressure on the public health system by preventing visits to emergency rooms and by reducing general practitioner engagements. Finally, yet importantly, a benefit to financial burden is expected [50].

### Study Limitations

Our study had some limitations. The first is the absence of a control group of unvaccinated patients or clinical data from the previous flu seasons for useful comparisons. Another evident limitation is the potential confounding effect of public health efforts to control COVID-19 that probably reduced influenza transmission. Indeed, the use of non-pharmaceutical interventions, such as isolation measures, mobility restriction, and public hygiene recommendations (i.e., hand-washing, face masks, gloves, etc.) prevented the circulation of different virus and bacterial pathogens [51]. Furthermore, cases of ILI were not confirmed by microbiological testing. Finally, given the relative low incidence of pneumococcal infection in cancer patients, the small sample size of our study prevented significance of the results.

## 5. Conclusions

Hospital-based influenza and pneumococcal vaccinations in patients under anticancer treatment and their FMs were successful in this single-center experience. Implementing in-hospital vaccinations for this population has the potential to increase vaccine coverage and to improve adherence, leveraging logistical convenience and the trust relationship with attending oncologists. Protecting patients with cancer from respiratory infections is of paramount importance. For this reason, we suggest providing in-hospital vaccinations for influenza and pneumococcal infections in the post-COVID-19 pandemic era as a reasonable approach to reduce the incidence and complications of respiratory tract infections.

## Figures and Tables

**Table 1 vaccines-12-00642-t001:** Demographic characteristics of the study populations.

	Patients n = 194	Family Members n = 147	*p*-Value *
Age, median (IQR)	63 (51, 7)	59 (29, 7)	0.02
Age > 65 years	87 (44.9%)	41 (33.9%)	0.07
Gender			<0.001
Male	63 (32.5%)	74 (50.3%)	
Female	131 (67.5%)	73 (49.7%)	

* Wilcoxon, chi-square, or Fisher’s exact test.

**Table 2 vaccines-12-00642-t002:** Post-vaccination outcomes and frequency of adverse events among patients and FMs.

	Patients n = 194	Family Members n = 147
ILI	18 (9.3%)	4 (2.7%)
PI	1 (0.5%)	1 (0.6%)
DoT	3 (1.5%)	NA
ED/HA	3 (1.6%)	0 (0.0%)
Local AE to I vaccine Missing	66 (34.0%) 0 (0.0%)	30 (21.6%) 8 (5.4%)
Local AE to P vaccine Missing	66 (37.7%) 0 (0.0%)	14 (20.6%) 8 (10.5%)
Overall systemic AE [I & P] ^†^ Missing	36 (20.6%) 0 (0.0%)	6 (8.3%) 4 (5.3%)
Overall systemic AE [I only] ^‡^ Missing	1 (5.3%) 0 (0.0%)	2 (2.9%) 3 (4.2%)

^†^ Only patients (n = 175) and family members (n = 76) who received both P and I vaccines were considered; ^‡^ Only patients (n = 19) and family members (n = 71) who received only I vaccine were considered. Abbreviations: FMs, family members; I, influenza; ILI, influenza-like illness; P, pneumococcal; PI, pneumococcal infection; DoT, delay of treatment; ED, visit to the emergency department; HA, hospital admission.

**Table 3 vaccines-12-00642-t003:** Outcomes of patients according to corticosteroid use.

	No Steroids (n = 64)	On Steroids (n = 130)	OR * (95% CI) Steroids vs. No Steroids	*p*-Value
Age, median (IQR)	63 (57,7)	63 (48,7)		
Age > 65 yrs	28 (43.8%)	59 (45.4%)		
Gender				
Male	17 (26.6%)	46 (35.4%)		
Female	47 (73.4%)	84 (64.6%)		
ILI	7 (10.9%)	11 (8.5%)	0.67 (0.24–1.87)	0.44
ED/HA	1 (1.6%)	2 (1.5%)	0.73 (0.06–8.68)	0.81
Local AE to I vaccine	20 (31.3%)	46 (35.4%)	1.15 (0.60–2.22)	0.67
Local AE to P vaccine Missing	22 (36.1%) 3 (4.7%)	44 (35.2%) 5 (3.8%)	0.95 (0.49–1.84)	0.87
Overall systemic AE	16 (25.0%)	21 (16.2%)	0.52 (0.24–1.13)	0.10

* Logistic regression analysis adjusted for age and gender. Abbreviations: ILI, influenza-like illness; ED, visit to the emergency department; HA, hospital admission; I, influenza; AE, adverse events, P, pneumococcal infection.

## Data Availability

Data are available upon request to the corresponding author. No online repository was used for ethical reasons.

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
