# Peer review of "Hospital-Based Influenza and Pneumococcal Vaccination for Cancer Patients on Active Treatment and Their Family Members during the COVID-19 Pandemic in Italy: A Single-Center Experience"

_vaccines, 2024, doi:10.3390/vaccines12060642_

Round 1

Reviewer 1 Report

Comments and Suggestions for Authors

I read the article with interest. I think it is interesting and deserves publication, the methodology is good and the results are clear. The limitations of the study are described.

In order to improve it, it should be considered that:

- there are few references, not only in absolute number but it would be interesting to see in the discussion part if there are other experiences at an international level on similar experiences.

- could this approach to vaccination for fragile/oncological subjects etc. be part of a wealth of knowledge and activities to be used in a pandemic? see for example: doi: 10.3390/healthcare9010017. you could highlight this aspect.

Author Response

Dear Editors,

We really thank the Reviewer 1 for his/her suggestions and his/her appreciation of our work.

Indeed, we enriched the bibliography with all the available significant previous experiences in the field.

In addition, we added the suggested article in the references and commented on the topic at the end of the discussion section, just before the "limitation" paragraph.

Reviewer 2 Report

Comments and Suggestions for Authors

The paper describes a population of patients (and family members) and the outcomes of the influenza vaccinations. The main results are that the vaccination reduces influenza like illness less in the patients than in family members. The paper is generally well written and on an important topic and deserves to be published.

The main concern is that the authors are trying to put a different spin on their results. On several places, they talk about vaccine hesitancy. They even have the following in the conclusions: "Implementing  in-hospital vaccination for this population has the potential to reduce vaccine hesitancy and improve adherence, leveraging on logistical convenience and trust relationship with attending oncologists. " Such conclusions are unjustified by the study design and presented results. There was nothing in the experiment that would allow the authors make claims about the hesitancy.

The minor concern is that there is a lot of detailed descriptions of the patients, such as Table 2. However, it is unclear if such a description is truly needed if not paired with the more detailed description of the outcomes (i.e. have only one table, merging Table 3 into Table 2 where the adverse outcomes would be recorded in the separate column and for each relevant category)

Author Response

Dear Editors,

We really thank the Reviewer 2 for his/her important considerations.

Indeed, we modified the manuscript by eliminating the previous statements regarding "vaccine hesitancy". Instead, we preferred to discuss "vaccine coverage" since we did not consider in detail all the required variables to analyze the "vaccine hesitancy".

Additionally, we merged Table 2 with Table 3 as suggested by Reviewer 2.

Reviewer 3 Report

Comments and Suggestions for Authors

Davide Dalu and colleagues (vaccines-3009336) presented a single center study of immunosuppression patients in morbidity and mortality of influenza and invasive Streptococcus pneumoniae infections, compared to the general population. As expected, vaccination is the most effective and safe approach to addressing the compelling questions, considering that such population is continuingly growing, and it is effective during the COVID-19 period. However, there are many other factors that could influence the respiratory disease burden during COVID-19, which are completely missing in the current version.

Major points:

1. During the COVID-19, there are many factors or interventions involved, while the famous one called COVID-19-related nonpharmaceutical interventions. This is very broad and sometimes fundamental impacts to many styles of infectious diseases, not only directly respiratory diseases, and disease of other forms, including both virus and bacteria. 

"While NPIs may have an indirect effect on bacterial transmission [3], improving hygiene practices, and reducing person-to-person contact [[4][5][6]], they may also affect the circulation of other pathogens, both directly and indirectly [7]. Previous studies have shown significant reductions or unusual patterns of shift in respiratory pathogens, such as respiratory syncytial virus (RSV) [8], influenza [8,9], and invasive pneumococcal disease (IPD) [10], predominantly respiratory pathogens [3]. Due to changes in human movement and associated behavior, certain diseases directly linked to personal transmission or travel contact, such as gastrointestinal, sexually transmitted, or vector-borne diseases, have been affected [11]. Other diseases, such as dengue fever [[12][13][14]], malaria [14,15], and amebiasis [16], have reduced cases due to NPIs associated with the international travel ban [9]." from doi: 10.1016/j.hlife.2024.03.005. 

This point is very important, and should be acknowledged in the text, with one paragraph.

2. Line 55-60, there are abnormal paragraphs; please formulate the idea and question with enough knowledge in the field.

3. Method (study design): please cut this part into several logic layers, it is really hard to follow the key points, please rewrite the guide with bullet points. Additionally, I would include a study design overview map to guide the reader.

4. Results

Very descriptive which is boring, I would suggest more informative figures and well-organized tables in the revised version, and more importantly, the authors should guide the text with the main findings as the subtitle in the manuscript.

5. Discussion,

as mentioned earlier, more critical point or paragraph should be included.

6. I would also suggest including a limitation paragraph for reader to fairly interpret the results and conclusion of the whole study. 

7. Finally, more informative literature should be included, there is a huge knowledge gap in this manuscript, which could not be put into the contextual studies in the field. This is very important for the reader in general.

Comments on the Quality of English Language

The language could be improved.

Author Response

Dear Editors,

We are profoundly grateful to Reviewer 3 for his/her fundamental suggestions that substantially improved the quality of the manuscript.

We added the suggested paper regarding the non-pharmaceutical interventions on COVID-19 to the bibliography and we decided to implement a comment on this important issue in the "limitation" section since these interventions could have impacted the results of our study.

As suggested, we modified the previous lines 55-60 and enriched the bibliography with all the significant knowledge derived from earlier studies in this field. We profoundly apologize for these missing pieces of information.

We tried to improve the "Study Design" section by following Reviewer 3 recommendations. Indeed, we divided the section into more understandable logic layers and used bullet points to help the reader follow the study flow. We did not add a study design overview map since this was an observational study with a relatively simple data collection and analysis method.

In addition, we added subtitles in the "Results" section to help the readers follow the data, as Reviewer 3 suggested. We avoided reorganizing Tables and adding Figures since the scope of our study was to enrich previous publications on the topic and to provide additional clinical information about the population of cancer patients receiving the study vaccines. The statistical analysis was limited to the differences in the outcomes between patients and family members, to confirm the patient population's increased frailty.

As suggested in point 6, we added a dedicated "limitation" paragraph.

Finally, we added a significant amount of articles to the bibliography, since there was an important gap of Knowledge, as highlighted by Reviewer 3. In this regard, we are considerably grateful to Reviewer 3.

Round 2

Reviewer 1 Report

Comments and Suggestions for Authors

the authors modified the text according to the suggestions

Reviewer 2 Report

Comments and Suggestions for Authors

The authors addressed my comments adequately and the paper is acceptable for publication.

Reviewer 3 Report

Comments and Suggestions for Authors

none